# Genome-Wide Association Mapping for Yield and Yield-Related Traits in Rice (*Oryza Sativa* L.) Using SNPs Markers

**DOI:** 10.3390/genes14051089

**Published:** 2023-05-15

**Authors:** Muhammad Ashfaq, Abdul Rasheed, Renshan Zhu, Muhammad Ali, Muhammad Arshad Javed, Alia Anwar, Javaria Tabassum, Shabnum Shaheen, Xianting Wu

**Affiliations:** 1Department of Plant Breeding and Genetics, Faculty of Agricultural Sciences, University of the Punjab, Lahore 54590, Pakistan; 2Department of Genetics, College of Life Sciences, Wuhan University, Wuhan 430072, China; 3Department of Entomology, Faculty of Agricultural Sciences, University of the Punjab, Lahore 54590, Pakistan; 4Department of Botany, Lahore College for Women University, Lahore 54590, Pakistan

**Keywords:** *Oryza sativa*, chromosomes, germplasm, GWAS, QTLs, SNPs, agronomic traits

## Abstract

Rice (*Oryza sativa* L.) is a staple food for more than 50% of the world’s population. Rice cultivar improvement is critical in order to feed the world’s growing population. Improving yield is one of the main aims of rice breeders. However, yield is a complex quantitative trait controlled by many genes. The presence of genetic diversity is the key factor to improve the yield hence, the presence of diversity in any germplasm is important for yield improvement. In the current study, the rice germplasm was collected from Pakistan and the United States of America and a panel of 100 diverse genotypes was utilized to identify important yield and yield-related traits. For this, a genome-wide association study (GWAS) was performed to identify the genetic loci related to yield. The GWAS on the diverse germplasm will lead to the identification of new genes which can be utilized in the breeding program for improvement of yield. For this reason, firstly, the germplasm was phenotypically evaluated in two growing seasons for yield and yield-related traits. The analysis of variance results showed significant differences among traits which showed the presence of diversity in the current germplasm. Secondly, the germplasm was also genotypically evaluated using 10K SNP. Genetic structure analysis showed the presence of four groups which showed that enough genetic diversity was present in the rice germplasm to be used for association mapping analysis. The results of GWAS identified 201 significant marker trait associations (MTAs. 16 MTAs were identified for plant height, 49 for days to flowering, three for days to maturity, four for tillers per plant, four for panicle length, eight for grains per panicle, 20 unfilled grains per panicle, 81 for seed setting %, four for thousand-grain weight, five for yield per plot and seven for yield per hectare. Apart from this, some pleiotropic loci were also identified. The results showed that panicle length (PL) and thousand-grain weight (TGW) were controlled by a pleiotropic locus OsGRb23906 on chromosome 1 at 10,116,371 cM. The loci OsGRb25803 and OsGRb15974 on chromosomes 4 and 8 at the position of 14,321,111 cM and 6,205,816 cM respectively, showed pleiotropic effects for seed setting % (SS) and unfilled grain per panicle (UG/P). A locus OsGRb09180 on chromosome 4 at 19,850,601 cM was significantly linked with SS and yield/ha. Furthermore, gene annotation was performed, and results indicated that the 190 candidate genes or QTLs that closely linked with studied traits. These candidate genes and novel significant markers could be useful in marker-assisted gene selection and QTL pyramiding to improve rice yield and the selection of potential parents, recombinants and MTAs which could be used in rice breeding programs to develop high-yielding rice varieties for sustainable food security.

## 1. Introduction

Rice (*Oryza sativa* L.) is a staple cereal crop that feeds more than 3 billion people of the world [1,2]. The major losses of rice yield production are due to a lack of diverse germplasm and various biotic and abiotic factors. Lack of genetic diversity limits crop production and poses a major problem in any breeding program for the development of new varieties. Thus, exploring natural diversity or creating genetic diversity (if not present) via different kinds of mutation is a critical step for the enhancement of yield. Rice breeders have used various kinds of germplasm like landraces, wild-type material, commercially approved varieties etc. to explore the genetic diversity within the existing germplasm to improve the yield. Yield is one of the most important and polygenic traits in rice. It is a complex and quantitative trait controlled by many genes and heavily influenced by environmental factors [3,4,5]. In rice, direct yield-determining traits are filled grains per panicle (G/P), panicle number per unit area and/or per plant (PNP), and TGW and indirect traits are plant height (PH), tillering ability (TA), growth period (GP), panicle length (PL), grains per panicle (GP), seed length (SL), and seed setting (SS) [6,7]. Therefore, the study of these traits is important for future rice breeding programs to improve grain yield. Fortunately, advancements in sequencing technologies provided ample opportunities than ever before to rapidly and efficiently dissect the genetic architecture of rice yield and yield components. The use of this genetic information provides plant breeders with a variety of modern tools and approaches to deeply dissect the genetic bases of yield and yield traits [8]. Therefore, developing new rice varieties with higher yields is crucial for future food security and is most challenging for rice breeders [9].

To efficiently use genetic diversity, it is useful to detect genomic regions governing the target trait so that marker-aided breeding can be employed. Biparental QTL mapping poses a limitation to explore the natural variation of the existing population. Alternatively, linkage disequilibrium (LD) based mapping, also known as genome-wide association study (GWAS), is an efficient and powerful strategy to utilize the natural variation of germplasm and helps in the identification of genomic loci. GWAS is a method of searching the entire genome for phenotypes showing continuous variation and mutations representing associations in natural individuals, or for lines using whole genome sequences [10]. Nowadays, the GWAS has become a prevalent study for the identification of QTLs. There are several advantages of GWAS over QTL mapping as it has the ability to identify the genes for a targeted trait in the natural population without deliberately creating it. A natural population has numerous recombination and GWAS has the potential to identify the causal mutations with higher resolution as compared to the QTL mapping. GWAS has recently become an ideally suited method for high-resolution mapping of agronomic traits in diverse rice populations [11]. Many useful traits including yield and yield-related traits and resistant loci-related biotic and abiotic stress tolerance has been mapped successfully using GWAS [12]. Therefore, LD-based mapping is a very powerful tool for the identification of important MTAs. The MTA study established the relationship among specific phenotypic and genetic variability within a genome, which ultimately detected the loci underpinning corresponding traits [13]. In the current study, the LD-based linkage mapping was applied to identify the yield-related traits in a diverse set of rice germplasm.

Single nucleotide polymorphism (SNP) is referred to as a difference of a single base in the sequence of DNA and the most general type of genetic marker to differentiate the individual organisms as well as traits. SNPs have become the fundamental genetic marker of selection for the analysis of completely or partially sequenced genomes because of their ubiquity within the genome. The large number of SNPs within the single genome can be utilized in the development of high-resolution genetic maps which can often lead to the linkage of SNPs with remarkable agronomic traits [14]. In the rice, the primary data of sequencing which led to the discovery of the first whole-genome, SNPs were derivatives from the draft cultivar sequences of the *japonica* and *indica*. The pools of SNPs were primarily restricted to measure the variability. However, with the advent of next-generation sequencing, the SNP markers are being utilized to map the important loci for many agronomically important traits [15,16,17,18]. However, re-sequencing costs and higher copies sequencing coverage are required for SNPs selection and map reconstructions of GWAS. Therefore, other methodologies should be explored such as; using fewer SNPs markers to construct fine maps for agronomic traits. This will reduce the budget and labor cost of data analysis procedures. Fewer SNPs could definitely map QTL loci better than traditional methods in the genetic population and the cost is far less than GWAS. Therefore, the main objective of this research is to map QTL for agronomic, morphological, and yield-related variables.

The results of association mapping presented in this study will be utilized for marker-assisted selection (MAS) on the same population.

## 2. Materials and Methods

### 2.1. Plant Material

In this study, a diverse set of rice germplasm containing 100 rice accessions was used (Appendix A). Seeds of the rice were obtained from the Rice Research Institute, Kala Shah Kaku and the United States Department of Agriculture (USDA), Washington, DC, USA.

### 2.2. Field Cultivation and Management

The selected rice germplasm were grown in the field area of the University of the Punjab Lahore, Pakistan (subtropical steppe/low-latitude semi-arid hot climate) for two growing seasons from 2016 to 2017 and 2017 to 2018. Seeds of each accession were soaked and sown in puddled field conditions from 15 to 20 April through the broadcasting method. After 25–30 days the nursery of each rice line was prepared for transplantation into the next field with three replications under a Randomized Complete Block Design (RCBD). Each block size was measured at about 16 m × 6 m (96 m^2^). Twenty plants of each accession were planted in each row and row to row distance was 30 cm and plant to plant distance was 15 cm. The soil type was silty clay. Based on the generally recommended dose, the fertilizer was applied after 30–40 days of transplanting the seedlings. NPK were applied with a ratio of 100:50:50 kg/ha. However, P and K were smeared basally after the transplant; N was smeared in three splits. Crop protection measures were practiced preventing pests and unwanted wild plants occasionally when necessary.

### 2.3. Phenotyping

At the time of maturity different agronomic traits were recorded i.e., Plant height (PH, in cm, the average of 5 plants of each line and replication), Days to 50% flowering (DTF, in days), Days to maturity (DM, in days), Tillers per plant (TP, the average of 5 plants of each line and replication), Panicle length (PL, in cm, the average of 5 plants), Number of grains per panicle (G/P, the average of 5 plants of each line and replication), Unfilled grains per panicle (UG/P, the average of 5 plants), Seed setting (%) (SS), the average of 5 plants of each line and replication), Thousand-grain weight (TGW, in g, the average of 5 plants of each line and replication), Yield kg/plot and Yield kg/ha. Details of the measurement procedures of each trait are given in (Appendix A).

### 2.4. DNA Isolation and Genotyping

Rice leaf samples were collected from 100 rice genotypes from the field and kept at −4 °C in the refrigerator for DNA extraction of each rice line. The DNA was extracted by the CTAB method using the protocol described earlier [19]. The quality of DNA was checked by running on gel electrophoresis and the quantity of DNA was checked on a NanoDrop 1000 Spectrophotometer (Thermo Scientific, Waltham, MA, USA). A 1.2 K multiplex-PCR panel, which was based on the GBTS platform from MolBreeding Biotechnol (http://www.molbreeding.com, accessed on 20 August 2017), was employed for genetic background analysis. The genome-wide positions of SNPs in terms of physical distance located on chromosomes were used in this study based on the Nipponbare reference sequence (RefSeq) (http://rapdb.dna.affrc.go.jp/, accessed on 20 August 2017).

### 2.5. Genotypic Data Analysis

Monomorphic markers, missing values < 20%, and shows unclear SNPs or (minor alleles) demonstrating allelic frequencies of less than 5%, were excluded from the analysis. Overall, 7098 out of the 10K functional iSelect beads chip analyses visually displayed polymorphic and were used for analysis. The mysterious relationships among individuals were calculated using a kinship matrix in the incorporated MLM [20].

### 2.6. Population Structure and GWAS Analysis

Bayesian clustering technique was applied with unlinked SNPs to classify groups of genotypically same individuals by applying the statistical software STRUCTURE v.2.3 [21]. Burn-in iterations of 104 cycles, followed by a simulation run of 106 cycles and the admixture model selection were used. The reliability and precision of the results were confirmed by running the K value at 10 runs for each K. The K value over the 10 runs was used to determine the most appropriate number of clusters [22]. We selected the K values ranging from 1 to 10 and 6 independent runs to attain reliable effects. The population structure was estimated by plotting the proposed number of subpopulations against the delta k [23]. 

GAPIT (genome association and prediction integrated tool) was also applied with the model selection preference to test the reliability of the results [24]. It was advanced in the R package which offers maximum likelihood precision and runs in a computationally effective method. (GAPIT) implements unconventional statistical approaches containing the compressed mixed linear model (CMLM) and CMLM-based genomic prediction and selection. Monomorphic markers, missing values < 20%, and shows unclear SNPs or (minor alleles) demonstrating allelic frequencies of less than 5%, were excluded from the analysis. The threshold for describing a marker to be significant was taken at 10^−4^ or above [25] after crossing the false discovery rate (FDR) at 0.05 value [24]. Overall, 7098 out of the 10K functional iSelect beads chip analyses visually displayed polymorphic and were used for analysis. To define the spurious associations derived from population structure, covariates from either STRUCTURE [22] or principal components (PCs) were considered as fixed effects. The mysterious relationships among individuals were calculated using a kinship matrix in the incorporated MLM [26]. The linkage disequilibrium (LD) in the studied diversity panel was evaluated using squared Pearson’s correlation coefficients (r^2^) using the r^2^ command in the software PLINK [27].

### 2.7. Gene Annotation

To determine the number of QTLs from all significant markers, the LD heat map R package was used to generate a graphical display of pair-wise linkage disequilibrium measures between SNPs in the genomic regions where significant SNPs (*p* < 1 × 10^−4^) were located. The QTL intervals were limited to regions where the R^2^ values (squared allele frequency correlation) between markers were above 0.4. In case the observed LD block around the significant marker(s) was less than 50 kb, we extended the QTLs up to 50 kb upstream and downstream of the detected regions. The SNP sequence was aligned to the rice genome through the blastn program with stringent E-values of 0.0001. For every SNP, only the best scoring hit was retained, and genomic position was annotated into 5′-UTR, 3′-UTR, CDS, intron, and intergenic regions conferring the genomic regions offered in the GFF3 files. The intergenic regions were distinct as genomic regions with no annotated genes. The annotated genes within ±250 Kb of the mapped SNP were considered candidate genes as described in a previous study [28,29].

### 2.8. Principal Component Analysis

Principal Component analysis was also used to determine phenotypic variability for these traits. phenotypic means were used for the PCA with respect to each trait. The result of PCA was used to make the clustering on phenotypic characters. The PCA was performed using IBM-SPSS statistics-20 [30].

## 3. Results

### 3.1. Phenotypic Diversity

The grain yield is one of the most important traits in rice crop improvement which is interlinked with other agronomic traits. The variation in the phenotypes is a key factor for crop improvement and an essential component of GWAS analysis. The trait-wise phenotypic variation was depicted by a boxplot (Figure 1). Significant differences in yield and yield-related traits were observed in the analysis of variance (ANOVA). Significant differences were observed in DF in season II and DM in both seasons and the remaining traits were highly significant in both seasons among studied rice germplasm (Appendix A). Across the rice genotypes, the mean value for the DF was 82.47 days in season I and 80.41 days in season II. The genotypes G58, G82, G83, and G84 flowered early in season I, the genotypes G56, G57, and G58 flowered early in season II, while G25 was found to be late flowering in both seasons. The DM were 110.65 and 108.95 days in season I and season II, respectively and ranged from 107 to 113 days in both seasons. The genotypes G36, G59, and G79 showed delayed maturity in season 1, while the genotypes G41 and G52 depicted delayed maturity in season II. Contrarily, an early maturity was observed in genotypes G68 and G72 in both seasons. The mean values for PH were 123.75 cm in season I and 121.33 cm in season II and ranged from 81.67cm to 145.33cm in both seasons. Genotypes G79 (145.33 cm) and G54 (143.3) were found to be taller genotypes G87 (81.67 cm) and G91 (79.63 cm) were dwarf. The TP was 13.43 in season I and 14.49 in season II. It ranged from 6.33 to 20.33 in season I and 7.39 to 22 in both seasons—the average PL was 24.86 cm in season I and 26 cm in season II. It ranged from 20.67 cm to 31.17 cm in both seasons. The G/P averaged 147.46 grains and 144.67 grains and it ranged from 85 grains to 204 grains in season I and from 82.94 grains to 202 grains in season II. The mean values for UG/P were 21 in season-I and 22.7 in season II. In season I, it ranged from 8.13 to 39.33 and in season II, from 9.82 to 42. The mean values for SS were 86.57 in season I and 88.4 in season II. It ranged from 63.78 to 99.29 in season I and 65.73 to 101.25 in season II. The mean value for TGW in the 100 rice genotypes were 26.95 g and 24.99 g in season I and season II, respectively. It ranged from 18.16 g to 32.1 g in season 1 and 16.21 g to 30 g in season II. The mean values for YP were 3.26 kg in season I and 3.33 kg in season II for this trait. In season I, it varied from 1.4 kg to 5.23 kg and from 1.49 kg to 5.32 kg in season II. The average Y/H across the studied rice genotypes was 2412.96 kg in season I and 2494.06 kg in season II. It ranged from 558.67 kg to 3893.33kg in season I and from 650.12 kg to 3984.79 kg in season II.

The principal components showed more variability with respect to eigenvalues (components having high eigenvalue corresponds high variability) that covered maximum variation among the traits. The PC1 showed 64% variation; the PC2 and PC3 showed 19% and 15% variation respectively and then finally decreased and stopped at 1% variation (Figure 2a). The PC1 is more related to TP, PH, DF, PL GP, UG/P, SS, TGW, MD and yield kg/plot. The PC2 showed positive effects with TP, PH, DF, SS, TGW and MD. The PC3 showed more relatedness with DF, PH, PL, UG/P and yield kg/plot. Similarly, the PC4 showed positive effects with TP, UG/P, TGW and yield kg/plot (Appendix A).

Overall, the Scree plot and scatter plot showed that the first four components are important (Figure 2b). This showed that for the studied attributes, a high level of phenotypic variability was observed among all accessions in the field condition. Phenotypic data showed a large variance in the studied rice diversity panel, making it useful for recording genotypic variability in the population. Based on the phenotypic analysis, genotype G13 showed the best performance in most of the yield and yield-related parameters and genotype G68 had the lowest performance as compared to other studied accessions.

### 3.2. Genotypic Diversity

The results of the web-based analysis showed the peak value at 4 (K = 4) indicating the presence of four subgroups among the studied rice accessions (Figure 3a). The results of the structure analysis also divided the germplasm into four groups which are depicted by four different colors (Figure 3b). Further assessment of each cluster or group revealed that genotypes G1–G10 and G27, G28 were found to fall in the first group; whereas the G11–G26 and G29, G33 genotypes completely appeared in the second group, G34–G72 fall in the third group and G73–G100 genotypes were grouped in fourth group (Table 1).

### 3.3. Genome-Wide Association Studies Using 10k SNP Array

Analysis of marker-trait association (MTA) was performed using the 10k SNP markers. The linkage disequilibrium (LD) patterns of different SNPs markers on all chromosomes are exhibited in (Figure 4a). Comparison between the marker density and the LD decay over distance indicates that markers are dense enough to have good coverage of LD. In the current study, LD decay over distance showed that the marker is dense enough for good coverage. Linkage disequilibrium is measured as R square for pair-wise markers and plotted against their distance. The moving averages of adjacent markers were calculated by using a sliding window with ten markers (Figure 4b).

A total of 201 significant MTAs were associated with studied traits, at or above −log 10 (*p* < 0.0001) threshold using a mixed linear model (MLM) for eleven yields and other agronomic traits (Appendix A). Marker trait associations were identified along with the desired phenotypic traits i.e., 16 MTAs were identified for plant height, 49 for days to flowering, three for days to maturity, four for tillers per plant, four for panicle length, eight for grains per panicle, 20 unfilled grains per panicle, 81 for seed setting %, four for thousand-grain weight, five for yield per plot and seven for yield per hectare. These MTAs are located on chromosomes 1, 2, 4, 5, 6, 7, 9, and 11 and explained the trait phenotypic variances ranging from 12.97% to 25.55%. Manhattan plots and quantile-quantile (Q-Q) plots generated through the model indicate that the model was well-fitted to data. The Manhattan plots show the location of significant SNPs and −log10 (*p*) associated with yield-related traits (Figure 5). Q-Q plots were used to assess the number and magnitude of observed associations between genotyped SNPs and traits under study. It was observed that p-values showed less deviation from the expected p-values. The blue horizontal line designates the threshold (*p* < 0.0001) of significance. The Y-axis in the (QQ) plot is the observed negative base 10 logarithm of the *P*-values, and the X-axis is the expected observed negative base 10 logarithms of the *p*-values under the assumption that the *p*-values follow a uniform [0, 1] distribution. The dotted lines shown in this plot are the 95% confidence interval for the QQ-plot under the null hypothesis of no association between the SNP and the studied traits as mentioned in. The *p*-values determine the association of QTLs with markers and R^2^ predicts the magnitude of QTL effects (Figure 6).

Most of the association signals were detected with close association to already identified regions. Details of trait and their associated SNP are described as follow:

#### 3.3.1. Tillers Per Plant (TP)

The TP was highly associated with four SNPs i.e., SNPs n OsGRb13190, OsGRg04446, OsGRg03402, and OsGRb29047 and they are located on chromosomes number 6, 3, 2, and 11, respectively (Figure 5). The phenotypic variation explained (PVE) for the trait ranged from 12.19% to 17.57% for the total phenotypic variation. The marker (OsGRb13190) explained maximum phenotypic trait variability (17.57%) on chromosome 1 at position 20,245,648 cM while the marker OsGRb29047 on chromosome 11 at position 858,733 cM explained minimum value (12.19%) in this study (Appendix A).

#### 3.3.2. Plant Height (PH)

The MTAs for PH were distributed across six chromosomes, including, eight MTAs at chromosome 7, four MTAs at chromosome 4, two MTAs at chromosome 4, and one MTA at each chromosome 1, 3 and 8 in this study. These MTAs explained from 11.47% to 25.55% of the total phenotypic variation (Appendix A). The SNP marker (OsGRb14446) explained maximum phenotypic trait variability (25.55%) on chromosome 7 at 14,594,194 cM while the marker OsGRb13785 on the same chromosome at 20,400 cM explained minimum value (11.47%). (Figure 5).

#### 3.3.3. Days to 50 Percent Flowering (DF)

A total of 49 SNPs were found to be located on chromosomes 2, 3, 7, 8, and 11 and significantly correlated with this trait (Appendix A). The PVE by the DF-associated markers exhibited from 15.05 % to 19.84% of the total phenotypic variation. The SNP (OsGRb30080) explained the maximum value of trait variability (19.84%) on chromosome 2 at position 18,212,087 cM while the SNP (OsGRb25578) from chromosome 3 at position 33,753,141 cM explained the minimum value (15.05%). (Figure 5). Marker trait association (MTAs) for DF were distributed across five chromosomes, including, 32 SNPs at chromosome 2, five SNPs at chromosome 3, nine MTAs at chromosome 11 and two SNPs at each chromosome 7 and 8 in the current study.

#### 3.3.4. Unfilled Grains per Panicle (UG/P)

A total of 20 significant MTAs were strongly linked with the UG/P. Overall, these significant SNPs were distributed across five chromosomes, Out of them, nine SNPs were on chromosome 8, four on chromosome 7, and three each on chromosomes 9 and 4, while, one SNP was on chromosome 5 (Appendix A). The phenotypic variation ranged from 13.17% to 15.74%. The SNP (OsGRg07442) on chromosome 5 at position 258,353 cM explained a maximum variation of 15.74% while; the SNP (OsGRg11693) on 8 at position 21,450,206 cM explained a minimum variation (13.17%) (Appendix A, Figure 5).

#### 3.3.5. Number of Grains per Panicle (G/P)

A total of eight significant MTAs were included, seven were located on chromosome 9 while one SNP was on chromosome 7. The phenotypic variation ranged from 23% to 20.79% (Appendix A). The marker (OsGRb28603) on chromosome 9 at position 12,952,275 cM explained a maximum variation of 20.79% while; the marker (OsGRb27733) on 4 at position 20,438,541 cM (Figure 5) explained a minimum variation (18.23%) for this trait.

#### 3.3.6. Days to Maturity (DM)

A total of three SNPs were strongly correlated with DM. Out of these, one SNP was from chromosome 3 (OsGRg04275), one from chromosome 4 (OsGRb09564), and one from chromosome 9 (OsGRb17703). The total PVE by these markers ranged from 11.57% to 12.97%. The SNP (OsGRb09564) had maximum PVE (12.97%) on chromosome 4 at 27,939,281 cM while the SNP (OsGRb17703) on chromosome 9 explained the least proportion of 6.53% of the trait variability at 11,125,119 cM position (Figure 5, Appendix A).

#### 3.3.7. Panicle Length (PL)

A total of four significant MTAs were found to be associated with PL. These MTAs for PL were distributed across the three chromosomes, including, two SNPs on chromosome 1 and one each on chromosomes 5, and 6. All of these SNPs explained phenotypic variation from 13.27% to 18.77% of the total phenotypic variability. The SNP marker (OsGRb23906) on chromosome 1 at position 101,16,371 cM explained maximum phenotypic variation (18.77%) while the marker (OsGRb10830) on 5 at position 10,350,550 cM explained minimum variation (13.27%) (Figure 5, Appendix A).

#### 3.3.8. Seed Setting Percentage (SS)

In total, 81 MTAs were found to be significantly associated with SS they were distributed across 5 chromosomes. Out of them, 71 were on chromosome 4, six on chromosome 8, two on chromosome 9, and one each on chromosomes 1, and 7. The PVE by the SS was from 16.33% to 22.69% of the total phenotypic variation. Chromosome-wise significant SNPs were OsGRb30591, OsGRb17738, OsGRg11329, OsGRg03752, and OsGRb14503 from chromosomes 4, 9, 8, 2, and 7 respectively exhibiting the maximum phenotypic variability (Figure 5, Appendix A).

#### 3.3.9. 1000 Grain Weight (TGW)

A total of four MTAs were strongly associated with TGW. Out of them, two were from chromosome 1 (OsGRb23906 and OsGRg01164) and one from each chromosome 2 (OsGRb05492) and 11 (OsGRb21690). Total phenotypic variation by these SNPs ranged from 13.77% to 15.63%. The marker (OsGRb23906) had maximum PVE, i.e., 15.63% on chromosome 1 at position 10,116,371 cM while the marker (OsGRb21690) on chromosome 11 at position 21,789,361 cM explained the least proportion i.e., 13.77% (Figure 5, Appendix A).

#### 3.3.10. Yield per Plot (Y/P)

A total of seven significant MTAs were found to be associated with Y/P. All the MTAs were distributed across five chromosomes, including, two SNPs each on chromosomes 4, and 7, and one SNP each on chromosomes 1, 8, and 9. The PVE ranged from 14.97% to 16.03%. The marker (OsGRb01011) on chromosome 1 at position 13,770,374 cM explained maximum phenotypic variation (16.03%) while the marker (OsGRg07137) on 4 at position 31,075,040 cM explained minimum variation (14.97 %) (Figure 5, Appendix A).

#### 3.3.11. Yield per Hectare (Y/H)

Yield per hectare was highly associated with five MTAs, out of which two were located on chromosome 4, while the others were on chromosomes 11, 3, and 12. All of these SNPs explained 17.37% to 18.71% of the phenotypic variation. The marker (OsGRb20658) explained maximum variation (18.71%) on chromosome 11 at position 7,220,561 cM while the marker (OsGRb23685) on chromosome 12 at 27,209,750 cM explained minimum variation (17.37%) (Figure 5, Appendix A).

Trait-wise highest numbers of MTAs were identified for SS (81) followed by DF (49), UG/P (20), PH (16), G/P (8), Y/P (7), Y/H (5) TG (4), PL (4), TP (4), DM (3). Chromosome-wise multi-trait-loci were perceived on chromosome 1 (PH, Y/P, TGW, and Y/H), 2 (DF, SS, TGW, and TP), 3 (DF, DM, PH, TP, and Y/H), 4 (DM, PH, SS, UG/P, Y/H, and Y/P), 5 (PL and UG/P), 6 (PH and TP), 7 (DF, G/P, PH, SS, UG/P, and Y/P), 8 (DF, PH, SS, UG/P, and Y/P), 9 (DM, G/P, SS, UG/P, and Y/P), 11 (DF, TGW, and TP) and 12 (Y/H) in this study. The MTAs detected herein provide an opportunity to clone genes and apply marker-assisted selection (MAS) under field conditions.

A pleiotropic locus is associated with and affects the expression of more than one phenotypic trait. In this study, several pleiotropic loci identified as pleiotropic locus OsGRb30591 on chromosome 4 at 12,914,840 cM were significantly linked with SS, UG/P and Y/H. Another pleiotropic locus OsGRb14503 on chromosome 7 at the position 15,365,358 cM was also associated with PH, SS, and UG/P. The studied traits like PL and TGW were controlled by a pleiotropic locus OsGRb23906 on chromosome 1 at 10,116,371 cM. The markers OsGRb25803 and OsGRb15974 on chromosomes 4 and 8 respectively, showed pleiotropic effects for SS and UG/P. A pleiotropic locus OsGRb09180 on chromosome 4 at position 19,850,601 cM was significantly linked with SS and Y/H in this study (Appendix A).

### 3.4. Gene Annotation of the Identified SNP Markers

On the Nipponbare reference genome (https://rapdb.dna.affrc.go.jp, accessed on 20 August 2017) of rice, 201 SNPs were successfully mapped which were found to be correlated with various attributes. In total 190 candidate genes were predicted around these MTAs. For DF, 49 candidate genes were predicted, three candidate genes for DM were predicted, and eight candidate genes for GP were predicted. A total of 16 and four candidate genes were found near SNPs associated with PH and TGW, respectively. For SS, 81 candidate genes were identified. For TP and UG/P 4 and 20 candidate genes were predicted, respectively. For GY/plot and yield per hectare, five candidate genes were predicted (Table 2, Appendix A). These genes need to be further analyzed for the regulation of certain pathways, and functional validation.

## 4. Discussion

For the efficient breeding program, information regarding the estimation of variability for the traits of genetic material is of utmost necessity. Therefore, phenotypic and genotypic components of variation would provide valuable information for the breeding of desirable traits [31,32]. The GWAS used both phenotypic and genotypic variations to dissect the complex traits and identify the functional genes related to diversity [33]. The same approach was applied in the current study. First, the phenotypic diversity of all yield and yield-related components was analyzed, and the result showed that enough phenotypic diversity was present in the current germplasm (Figure 1). This indicates that genotypes are highly variable, especially those traits which showed significant differences. Thus, the possible genetic improvement through selection is highly promising. In this study, genotype G13 showed the best performance in most of the yield and yield-related parameters and genotype G68 had the lowest performance as compared to other studied accessions. Similar results were obtained in earlier studies [34,35]. Similarly, phenotypic diversity was also evaluated through principle component analysis (PCA) and the result showed that the maximum variation is covered in four PCs. Along with the phenotypic diversity, the genotypic diversity was elucidated using STRUCTURE analysis and it was found that four types of subpopulations were found in the current germplasm (Figure 3). Based on these results it is predicted that current germplasm is diverse enough to be used in any breeding programs.

Agronomic traits are highly associated with rice production and serve as key factors for grain yield and its consumption and market value [36]. The variation in agronomic traits can be identified and traced back to the underlying causative loci via various mapping approaches which include quantitative trait locus (QTL) mapping and GWAS. In this perspective, a GWAS tool was applied to connect the phenotypic variation to genotype. GWAS is a beneficial tool for recognizing the positions of genes, QTLs, and candidate genes liable for the variations in the desired quantitative characters [37]. In the recent era, the GWAS is being successfully utilized to dissect the complex traits in rice, which includes both quantitative and qualitative traits [38,39]. For example, a study identified 80 MTAs for GY, PH and DM on chromosomes 2, 5, 10, 11, and 12 [40]. Another study identified 43 SNPs which are associated with yield and yield-related traits [41]. Moreover, 2255 MTAs signals were detected for yield and its related traits, and the significant SNPs were distributed in 903 genes [42]. Similarly, in the current study, 10k SNP data was utilized to find the diversity as well as to find the significant associations with phenotypes. In this study, GWAS was performed using high-quality sequencing data from a 100-rice core collection and phenotypic data, with determined the significance. It was confirmed that significant SNPs were detected based on −log10 (*p*) > 3 in a Manhattan plot, according to the yield and yield-related traits. In addition, the expected p-value of the x-axis and observed p-value from significant SNPs showed a diagonal linear shape in the QQ plot, which means that the discovered SNPs expressed their characteristics well, with normality and significance.

The results showed that 201 MTA were significantly associated with yield parameters. The MTAs detected herein will be helpful in cloning the genes and this information will be applied for marker-assisted selection (MAS) for yield improvement and dissecting the genetic mechanism of important cultivars in rice. Therefore, MTAs identified in the current study are vital because these could be associated with minor genes related to the targeted trait. The SNP markers, associated with targeted loci can be used for pyramiding favorable alleles in newly developing varieties with improved traits. Moreover, these varieties could be potentially used as parents in breeding programs for the genetic improvement of grain yield.

The phenomenon of pleiotropy will occur when a gene is controlling more than one phenotypic trait [43,44,45]. There are two distinct but overlapping mechanisms of pleiotropy, one is gene pleiotropy and the second is region pleiotropy [46]. Like gene pleiotropy, region pleiotropy occurs when a certain region is linked with two or more traits. The region pleiotropy is divided into two categories based on their functions; one is known as unfavorable linkage or linkage drag, and the other is known as favorable linkage [46]. In the GWAS studies, identification of the favorable pleiotropic loci for grain yield is desired. With the development of sequencing technologies, SNP-based GWAS has become an effective method to detect the favorable pleiotropic regions across the whole genome [47]. Using the GWAS studies, several pleiotropic loci were identified such as six pleiotropic regions on chromosomes 1, 2, 3, 5, and six were identified in rice controlling two or more determinants of rice grain [48]. Similarly, another study identified several pleiotropic regions associated with grain width and grain length-to-width ratio in rice [49]. Parallel studies were conducted in other experiments and multi-trait loci were identified for yield and yield-related traits in crop plants [50,51,52,53]. Similarly in the current study, several pleiotropic loci were identified such as pleiotropic locus OsGRb30591 were significantly linked with SS, UG/P and Y/H. Another pleiotropic locus OsGRb14503 was associated with PH, SS, and UG/P. the pleiotropic locus OsGRb23906 controls PL and TGW (Appendix A). The major-effect of these significant markers is useful in marker-assisted selection and QTL pyramiding to improve rice yield. These above advances confirm that GWAS could be an effective means to identify significantly associated SNPs and candidate genes associated with the studied traits in rice. This comprehensive study provides a timely and important genomic resource for breeding high yielding rice genotypes.

## 5. Conclusions

Phenotypic and genotypic diversity is crucial for any breeding program. In the current study, the diverse set of 100 rice germplasm was assessed with respect to its phenotypic diversity with PCA. The results divided the whole germplasm into four groups. Similarly, genetic STRUCTURE results divided the germplasm into four groups based on the genotypic similarities and dissimilarities. The GWAS identified 201 significantly associated MTAs with 11 important agronomic traits. In contrast, gene annotation results identified 190 candidate genes having close and significant associations with yield and yield-related traits. For DF, 49 candidate genes were predicted, three candidate genes for DM were predicted, eight candidate genes for GP were predicted. A total of 16 and four candidate genes were found near SNPs associated with PH and TGW respectively. For SS, 81 candidate genes were identified. For TP and UG/P, four and 20 candidate genes were predicted, respectively. For YP and YH, five candidate genes were predicted for each trait in the current study. The identified candidate genes and associated markers in this study enable efficient testing of cross populations and provide materials that can be applied in rice breeding programs. These results could be very useful for the selection/screening of potential parents and further developments of new high-yielding recombinants in the rice field.

## Figures and Tables

**Figure 1 genes-14-01089-f001:**
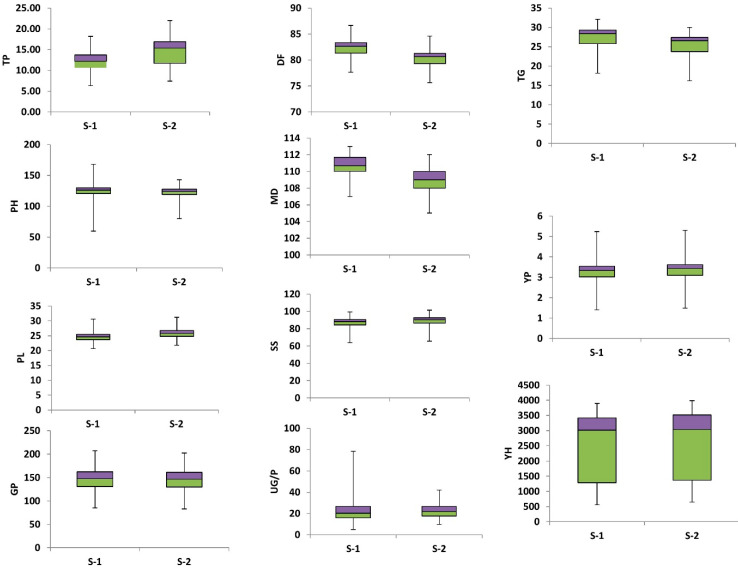
Boxplots of 11 agronomic traits within diverse sets of rice germplasm. Number of tillers per plant (TP), Plant height (PH), Panicle length (PL), Number of grains per panicle (G/P) Days to 50 percent flowering (DF), Days to maturity (DM), Number of unfilled grains per panicle (UG/P), Seed setting percentage (SS), 1000 grain weight (TGW), Yield per plot (Y/P) and Yield per hectare (Y/H). S-1 and S-2 represent season 1 and season 2. Bars represent the minimum and maximum values of the target trait.

**Figure 2 genes-14-01089-f002:**
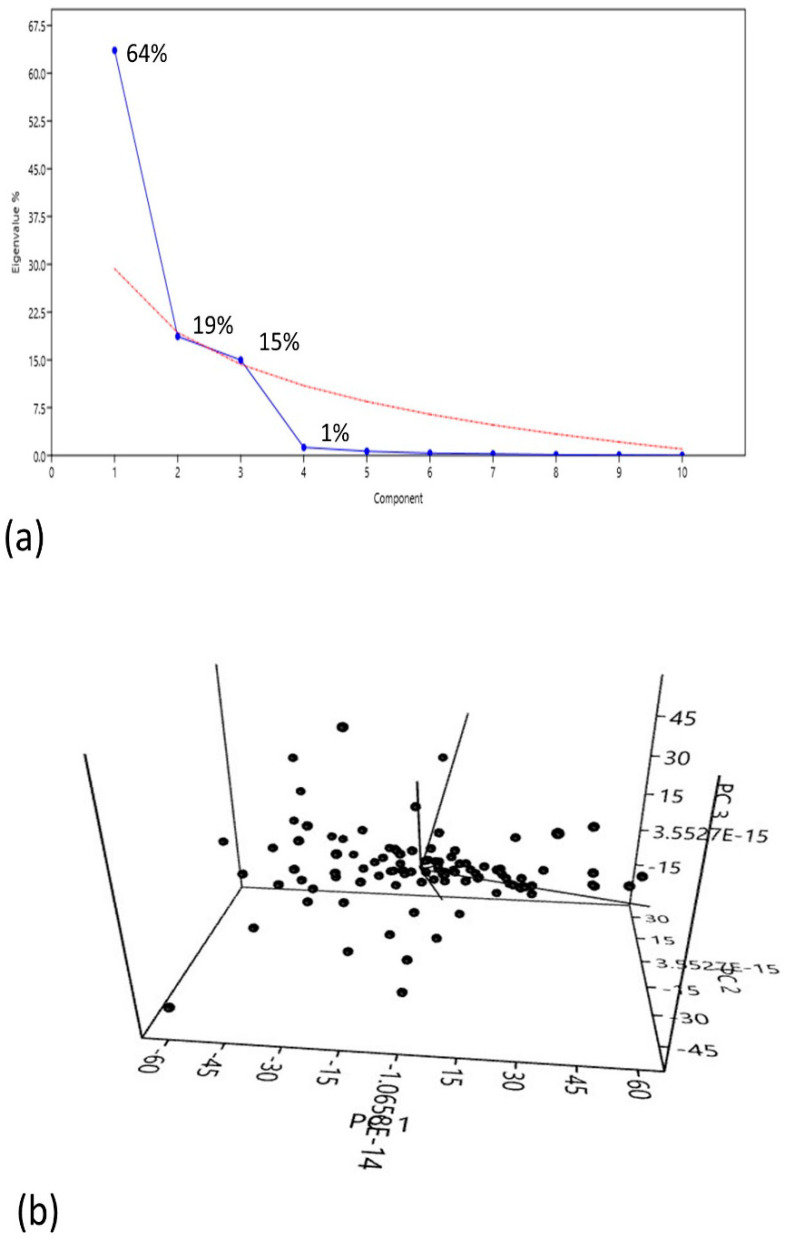
Principal component analysis (PCA) of 11 agronomic traits, (**a**) scree plot represents the number of components in which maximum variation is covered. The variation in components is represented in term percentages. (**b**) Variation captured by the PCs showing the clustered distribution along the first three principal components.

**Figure 3 genes-14-01089-f003:**
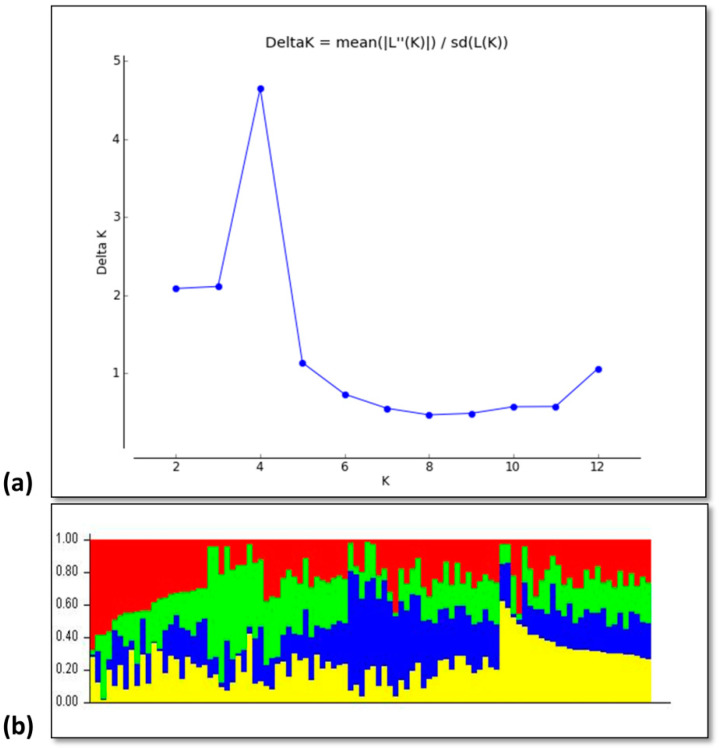
The population STRUCTURE of 100 rice accessions. (**a**) Prediction of the number of groups present in the germplasm based on Q-matrix at k = 10. (**b**) The genetic relatedness of diverse rice germplasm based on STRUCTURE analysis representing the presence of four groups in rice germplasm. Four different colors represent the presence of four groups in rice germplasm.

**Figure 4 genes-14-01089-f004:**
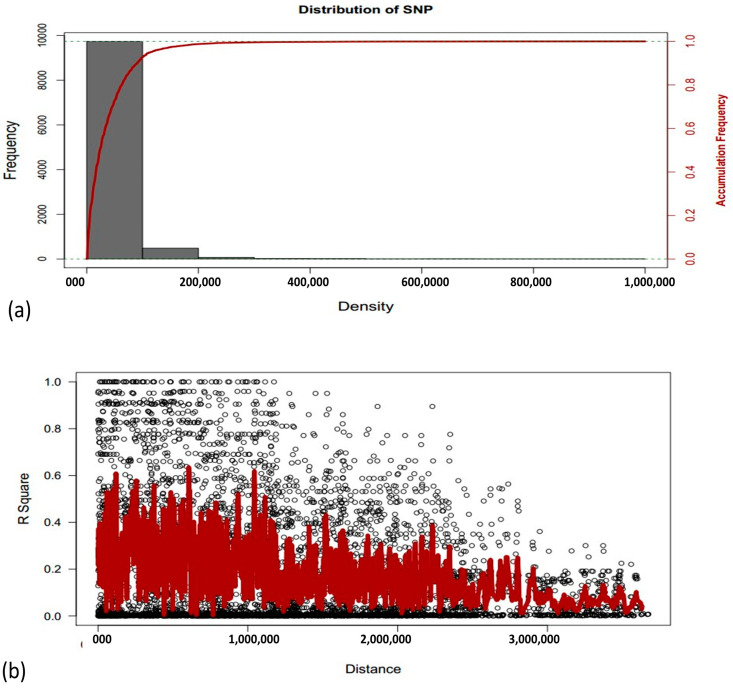
(**a**) Frequency and accumulative frequency of marker density. (**b**) chromosomes-wise linkage disequilibrium (LD) decay over distance. Each dot represents a pair of distances between two markers on the window and their squared correlation coefficient. The red line is the moving average of the 10 adjacent markers.

**Figure 5 genes-14-01089-f005:**
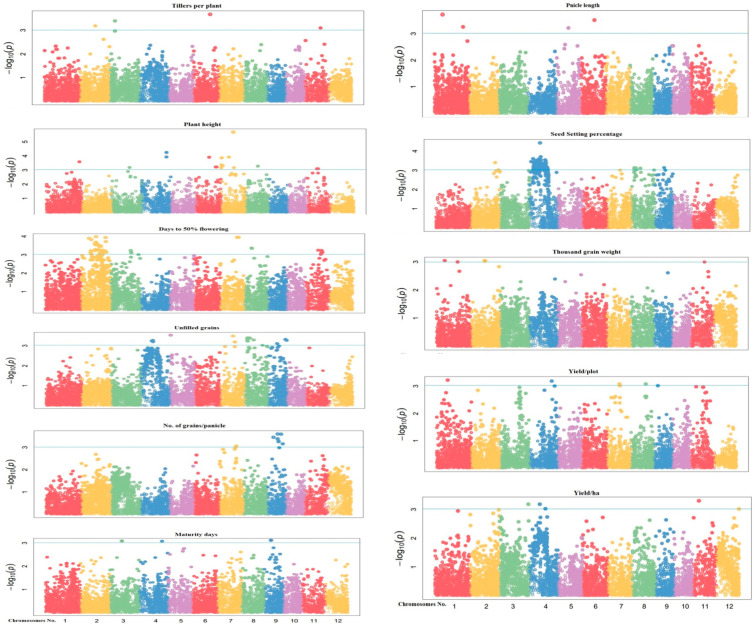
Manhattan plots of 11 agronomic traits representing the identification of SNP markers: Manhattan plots of −log10 (*p*-value) versus chromosomal position of SNP markers associated with the different agronomic traits of rice. The x-axis represents chromosomal locations and the y-axis, −log10 (*p*-values) from genotypic associations. The blue horizontal line represents the genome-wide adjusted FDR correction log10 (*p*-value = 3).

**Figure 6 genes-14-01089-f006:**
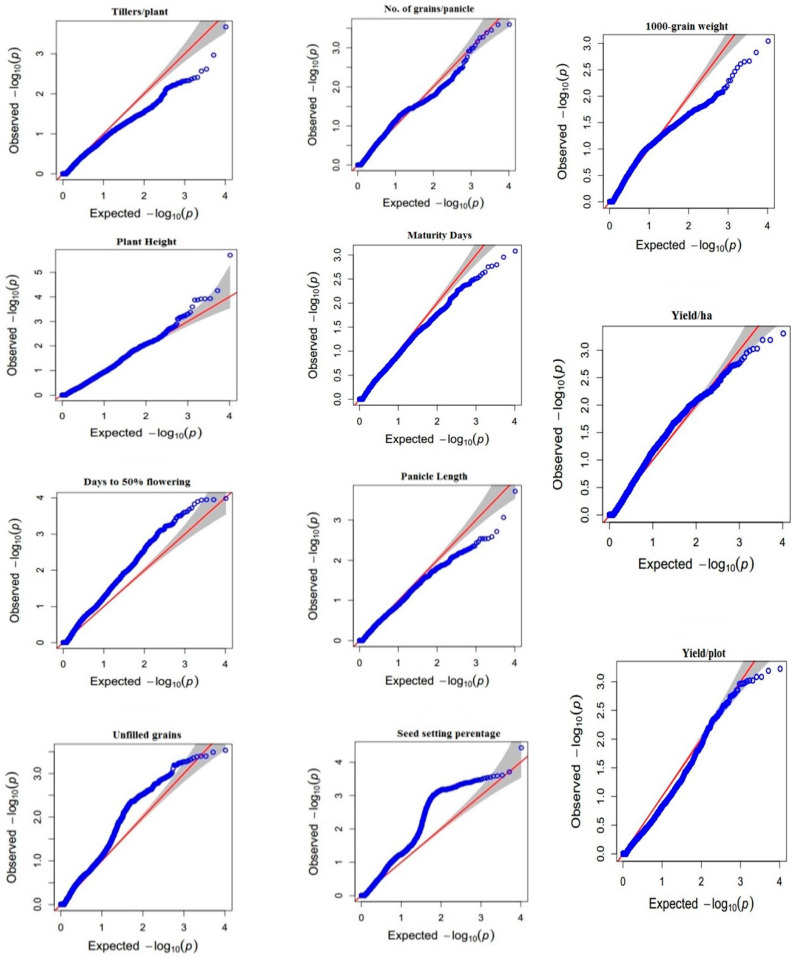
Q-Q plots of 11 agronomic traits representing the expected and observed values. The red line represents the expected p values while the blue lines show the deviation from expected values.

**Table 1 genes-14-01089-t001:** Distribution of genotypes/varieties in different groups based on STRUCTURE analysis.

S. No	Group	Number of Genotypes	Name of Genotypes
1	Group I	12	Lemont?, Bond, IR-36, Delvex, Teqing, Nira, Cica, Koshihikari, IR-64, Bellmont, Taducan, Dee Geo Woo
2	Group II	21	Yangzi-95, Gui-99, L-203, CE-65, Zao-40, LA-110, Rando, Jakson, WC-4644, Tsai Yuan Chung, Cica-6, IR-456-3-2-1, Newbonnet, Newrex, Taichung Native-1, Sinum Paga Selection, Stg-663228, Lebonnet, Starbonnet, Della, Toro-2
3	Group III	39	Delitus, Dellrose, CDR-448, CDR-201, B5-Xiequizao, Roxero regue, H-256-76-1-1-1, Palman, Jasmine-85, A-301, L-202, VE GOLD, L-203, IR-6, Sathi basmati, shaheen basmati, basmati-198, basmati-370, basmati-Pak, Basmati-385, Basmati-515, R- 456, CB-5, CB-10, CB-11, CB-12, CB-13, L-203, VeGold, TP-49, Hill Long Grain, L-202, A-301, L-202, V-203, PALMAN, 87-1-550, 79, 923
4	Group IV	28	CB-14, CB-15, CB-16, CB-17, CB-19, CB-20, CB-209, CB-21, CB-22, CB-26, CB-27, CB-28, CB-29, CB-30, CB-31, CB-32, CB-33, CB-34, CB-36, CB-38, CB-39, CB-40, CB-41, CB-43, CB-44, KSK-282, KSK-133, Roxero regue

**Table 2 genes-14-01089-t002:** Details of important SNP markers identified for yield and yield-related traits along with total MTAs.

Sr. No	Traits	SNP	Chro	Position	Gene ID	Region	*p* Value	R^2^	Strand	MTAs
1	PH	OsGRb14446	7	14594194	Os07g0436100|13882; Os07g0436350|3483; Os07g0437000|24064	Intergenic	2.02 × 10^−6^	25.55	−	16
2	DF	OsGRb30080	2	18212087	Os02g0508500|45221; Os02g0510100|19402; Os02g0510300|27182	Intergenic	1.03 × 10^−4^	19.84	+	49
3	DM	OsGRb09564	4	27939281	Os04g0557500	CDS	8.27 × 10^−4^	12.97	−	3
4	T/P	OsGRb13190	6	20245648	Os06g0538900|20956; Os06g0539100|13974; Os06g0539500|11602; Os06g0540050|35029; Os06g0540200|36824	Intergenic	2.11 × 10^−4^	17.57	−	4
5	PL	OsGRb23906	1	10116371	Os01g0283000; Os01g0283000	Intron	1.91 × 10^−4^	18.77	−	4
6	G/P	OsGRb28603	9	12952275	Os09g0381600|37451	Intergenic	2.52 × 10^−4^	20.79	+	8
7	UG/P	OsGRg07442	5	258353	Os05g0104700	3UTR	2.92 × 10^−4^	15.74	+	20
8	SS	OsGRb30591	4	12914840	Os04g0294401|3101; Os04g0294812|20440; Os04g0295100|40925	Intergenic	3.70 × 10^−5^	22.69	−	81
9	TGW	OsGRb23906	1	10116371	Os01g0283000; Os01g0283000	Intron	9.02 × 10^−4^	15.63	−	4
10	Y/Plot	OsGRb01011	1	13770374	Os01g0346700|8390; Os01g0347100|29581; Os01g0347200|37151	Intergenic	5.94 × 10^−4^	16.03	−	7
11	Y/H	OsGRb20658	11	7220561	Os11g0235250|23265; Os11g0235700|1114	Intergenic	4.95 × 10^−4^	18.71	+	5

Days to 50 percent flowering (DF), Days to maturity (DM), Plant height (PH), Number of tillers per plant (TP), Panicle length (PL), Number of grains per panicle (G/P), Number of unfilled grains per panicle (UG/P), Seed setting percentage (SS), 1000 grain weight (TGW), Yield per plot (Y/P) and Yield per hectare (Y/H). “+” represent the forward strand and “−” represent the reverse/complementary strand of DNA.

## Data Availability

The data that support the findings of this are available in the main manuscript and the Appendix A.

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
