# Peer review of "Genome-Wide Association Mapping for Yield and Yield-Related Traits in Rice (Oryza Sativa L.) Using SNPs Markers"

_genes, 2023, doi:10.3390/genes14051089_

Round 1

Reviewer 1 Report

The authors have made great efforts to analyze the significant associated SNPs based on 11 important agronomic traits from 100 rice germplasms. The attained data are of interest to the readers and agronomists to develop the effective high-yield rice breeding program. However, the current manuscript has not been well organized and documented and needs critically revised and edited to improve its quality of the manuscript. The authors can find some useful comments and suggestions below:

1. Abstract must be reworded and should cover the brief backgrounds of the research topic, method and significant results as well as the conclusion.

2. Introduction, some information needs to confirm, for example, lines 82-86, "using less (fewer) SNPs to construct a less fine map for agronomic trait-related QTL...." which is possibly leading to gain precise results? 

3. At the end of the introduction part, should clearly narrate the objectives of this study.

4. In Materials and Methods, as shown in Sup Table 1, why did the authors know these lines are indica or japonica etc?

5. The term "rice germplasms, rice germplasm lines, rice accessions" should be made uniformity

6. subsection 2.2 " Field cultivation and management" the field location must be added more information, such as names of areas, and total square areas for experiments, soil components, climate information etc.,  are there any differences between soil components between two growing seasons? because the environmental factors may affect the agronomic traits of rice

7. Lines 177-179, the authors mentioned "SNP were considered as candidate as..." what does it mean? provide specific the previous study? and the term "candidate" or candidate gene" must be mentioned in the introduction part.

8. Part 2.8 "Principal component analysis" must be rewritten, because many agronomic traits were analyzed in this study, hence statistical analyses must be clearly described.

9. Results: Some parts are confusing and should be judiciously revised. Fig 1 should be rearranged, for example, in line 252-253, "Yield per plot" should be mentioned in Materials and Methods, even though, the authors have already stated in Sup material

10. Part 3.2 "Genotypic diversity" should be reworded, no need to repeat the method in this section

11. Fig 3 a, is not good and should be replaced 

12. Discussion, in my opinion, this part is no need to divide into 2 subsections, join in one. This part should be rewritten and deeply discussed the significantly attained results of this study, compared and discuss to other related worldwide studies, 

13. English writing in this paper has not yet reached the published level, hence needs to be critically edited for both spelling and grammar errors.

Author Response

Dear Editors of Genes, MDPI,

Thank you very much for your constructive comments regarding our manuscript entitled "Quick genome-wide screening for yield and yield related traits in rice (Oryza Sativa L.) using SNPs markers”. We have carefully revised the manuscript according to the suggestions. Our point-by-point responses to the Reviewers comments are presented below.

Reviewer 1:

The authors have made great efforts to analyze the significant associated SNPs based on 11 important agronomic traits from 100 rice germplasms. The attained data are of interest to the readers and agronomists to develop the effective high-yield rice breeding program. However, the current manuscript has not been well organized and documented and needs critically revised and edited to improve its quality of the manuscript. The authors can find some useful comments and suggestions below:

  1. Abstract must be reworded and should cover the brief backgrounds of the research topic, method and significant results as well as the conclusion.

Answer: Thanks for the nice suggestion. The Abstract is reworded now. Please check the changes in the track change system from line 12-41.

  1. Introduction, some information needs to confirm, for example, lines 82-86, "using less (fewer) SNPs to construct a less fine map for agronomic trait-related QTL...." which is possibly leading to gain precise results? 

Answer: Thanks to point out the writing errors. We have carefully revise the sentences and overall the introduction section. Please in track changes mode as “This will reduce the budget and labour cost on data analysis procedures. Less number of SNP could definitely map QTL loci better than traditional methods in the genetic population and the cost is far more less than GWAS”.

  1. At the end of the introduction part, should clearly narrate the objectives of this study.

Answer: Thank you for the suggestion. In the end of the introduction part we have clearly narrates the objective of the study. Please see line 104-109 in revised manuscript

  1. In Materials and Methods, as shown in Sup Table 1, why did the authors know these lines are indica or japonica etc?

Answer: Thanks for the nice inquiry. The Indica and Japonica type are subspecies of O.sativa. The studied germplasm belong to the Indica type. Both of these subspecies have their own characteristic like japonica grains are short, roundish and its grains do not shatter easily. While indica rice have a long, slender and somehow flat grains and shatter easily.

  1. The term "rice germplasms, rice germplasm lines, rice accessions" should be made uniformity.

Answer: Thanks for making the clarity for readers. We have carefully revised the manuscript and use the term rice germplasm throughout the manuscript.

  1. subsection 2.2 " Field cultivation and management" the field location must be added more information, such as names of areas, and total square areas for experiments, soil components, climate information etc.,  are there any differences between soil components between two growing seasons? because the environmental factors may affect the agronomic traits of rice

Answer: Thanks for the nice comment. We have added the relevant information in field cultivation and management section. Please see lines 145-155there is no difference in soil components during both seasons. Although the environmental changes exits as it occurs naturally.

  1. Lines 177-179, the authors mentioned "SNP were considered as candidate as..." what does it mean? provide specific the previous study? and the term "candidate" or candidate gene" must be mentioned in the introduction part.

Answer: Thanks for the nice inquiry. The term MTA is different from candidate genes. The MTA (marker trait association) means associated SNP with the trait of interest. While during gene annotation it is search for the predicted gene that is around/within that associated SNP marker. That predicted gene is known as the candidate gene. The functions for the candidate genes need to be validated.  The study about the MTAs were already was mentioned in the introduction section.

  1. Part 2.8 "Principal component analysis" must be rewritten, because many agronomic traits were analysed in this study, hence statistical analyses must be clearly described.

Answer: Thanks for the suggestion. The statistical analysis is already described in material and method section. Please see line 227-229

  1. Results: Some parts are confusing and should be judiciously revised. Fig 1 should be rearranged, for example, in line 252-253, "Yield per plot" should be mentioned in Materials and Methods, even though, the authors have already stated in Sup material

Answer: Thanks for your inquiry. The yield per plot is already mentioned in the material and method section. Please see line 166-167.

  1. Part 3.2 "Genotypic diversity" should be reworded, no need to repeat the method in this section.

Answer: Thanks for useful comment. We have reworded the results described in the part 3.2. Please see the line 318-329.

  1. Fig 3 a, is not good and should be replaced 

Answer: Thanks for good suggestion. We have replaced the Figure 3a with high quality image.

  1. Discussion, in my opinion, this part is no need to divide into 2 subsections, join in one. This part should be rewritten and deeply discussed the significantly attained results of this study, compared and discuss to other related worldwide studies, 

Answer: Thanks for the critical evaluation. We have modified the discussion section according to your suggestions.

  1. English writing in this paper has not yet reached the published level, hence needs to be critically edited for both spelling and grammar errors.

Answer:  Thanks for the critical comment. We have carefully revised the manuscript with respect to the English and grammar error.

Thank you very much for your time and consideration.

Sincerely

Dr. Muhammad Ashfaq,

Associate Professor

Department of Plant Breeding & Genetics,

Faculty of Agricultural Sciences, University of the Punjab, Lahore.

Reviewer 2 Report

Dear Authors,

An interesting study that attempts to map genes yield and important agronomic traits of rice using genome-wide screening. The introduction needed to be adequately articulated the problem statement, and clearly defined objectives needed to be included. Overall, it could be better written and there is plenty of scope to bring clarity so that this important research can be reproduced.

Line 138: Please consider adding details on how you discovered SNPs, what were the initial number of SNPs before filtering and after removing monomorphic, missing, minor alleles what numbers were derived. Finally, how many SNPs were used in the downstream analysis.

Line 158: The threshold seems too flexible and subjected to false marker trait association because these SNPs are not independent due to linkage disequilibrium. To restrict false marker trait association, you can consider applying Li and Ji (2005) threshold (Heredity 95 (3): 221–227).

I made several comments and suggestions throughout the text. Please check those carefully and implement those throughout the text beyond the comment location.

Thank you,

Reviewer

Author Response

Dear Editors of Genes, MDPI,

Thank you very much for your constructive comments regarding our manuscript entitled "Quick genome-wide screening for yield and yield related traits in rice (Oryza Sativa L.) using SNPs markers”. We have carefully revised the manuscript according to the suggestions. Our point-by-point responses to the Reviewers comments are presented below.

Reviewer 2:

Dear Authors,

  1. An interesting study that attempts to map genes yield and important agronomic traits of rice using genome-wide screening. The introduction needed to be adequately articulated the problem statement, and clearly defined objectives needed to be included. Overall, it could be better written and there is plenty of scope to bring clarity so that this important research can be reproduced.

Answer: Thanks for the critical review. We have modified/ rewritten the introduction section articulating the problem statement. Please see the entire revise introduction in track changes.

  1. Line 138: Please consider adding details on how you discovered SNPs, what were the initial number of SNPs before filtering and after removing monomorphic, missing, minor alleles what numbers were derived. Finally, how many SNPs were used in the downstream analysis.

Answer: The genome-wide positions of SNPs in terms of physical distance located on chromosomes were used in this study based on the Nipponbare reference sequence (RefSeq) (http://rapdb.dna.affrc.go.jp/). Monomorphic markers, missing values< 20%, and shows unclear SNPs or (minor alleles) demonstrated the allelic frequencies of less than 5%, were excluded from the analysis. Overall, 7,098 out 10K functional iSelect beads chip analyses visually displayed polymorphic and were used for analysis

  1. Line 158: The threshold seems too flexible and subjected to false marker trait association because these SNPs are not independent due to linkage disequilibrium. To restrict false marker trait association, you can consider applying Li and Ji (2005) threshold (Heredity 95 (3): 221–227).

Answer: Thanks for highlighted the main point. The threshold for describing a marker to be significant was taken at 10-4 or above after crossing the false discovery rate (FDR) at 0.05 value to avoid the False marker trait association and these types of association were not included in GWAS results.

  1. I made several comments and suggestions throughout the text. Please check those carefully and implement those throughout the text beyond the comment location.

Answer: we have carefully revised the comments. Our point by point answer is as below:

  1. Why it named Quick?

Answer: thanks for the inquiry. As compare to QTL mapping Association study reduce the time and labour for making the population and crossing efforts. That’s why it is quick method for mapping the genetic loci.

  1. Are sequencing technology belongs to biotechnological tools? Rather try 'Advancements in sequencing technology provided ample opportunities to rapidly and efficiently dissect the genetic architecture of rice yield and yield components more than ever before.

Answer: Thanks for the nice suggestion. We have revised the sentence. Please see line 62-69

  • What were the bases for determining this conceptual population of 100 rice accession.

Answer: Thanks for the nice inquiry. Our main goal is to collect the as many accessions as we can and we maximally get the 100 accession for this study.

  1. Please add further details on the genotyping dataset, insitial SNPs and futher screening derived SNPs like after filtering for missing values, MAF, heterozygosity etc...

Answer: The genome-wide positions of SNPs in terms of physical distance located on chromosomes were used in this study based on the Nipponbare reference sequence (RefSeq) (http://rapdb.dna.affrc.go.jp/). Monomorphic markers, missing values< 20%, and shows unclear SNPs or (minor alleles) demonstrated the allelic frequencies of less than 5%, were excluded from the analysis. Overall, 7,098 out of the 10K functional iSelect beads chip analyses visually displayed polymorphic and were used for analysis.

  1. Line 162- How many PCs were used?

Answer: Thanks for inquiry. In this study 4 PCs were used which have more variations as compared to others

  1. I think it would be a good idea to add heritability estimates for the assessed traits. The estimates would give an idea about the genetic architecture of the traits, and a validation for GWAS results

Answer: Thanks for nice suggestions. For the validation of GWAS results, the population structure and false discovery rate was calculated. In our opinion if we add heritability it will change the entire results of GWAS. It’s not always necessary to calculate the heritability to validate the GWAS results. There are many papers without the heritability calculation. Please see (https://doi.org/10.1186/s12284-020-00431-2). So, it is requested that please consider our previous results.

  • Figure 4. The LD decay line is not clear. I suggest reproducing the graph to clearly exhibit genome wide LD decay and LD decay to half. Please do not just put from GAPIT output.

Answer: Thanks for critical review. The QTL intervals were limited to regions where the r2 values (squared allele frequency correlation) between markers were above 0.4 (P < 1e-04). In case the observed LD block around significant marker(s) was less than 50 kb, we extended the QTLs up to 50 kb upstream and downstream of the detected regions. In this study, GWAS result obtained from GAPIT and Figure 4 are again generated, and their detail included in the revised manuscript like “Each dot represents a pair of distances between two markers on the window and their squared correlation coefficient. The red line is the moving average of the 10 adjacent markers.

  • Line 302- P=0.00001. the threshold is too flexible, you can try applying Bonferroni or similar correction.

Answer: The threshold for describing a marker to be significant was taken at 10-4 or above  after crossing the false discovery rate (FDR) at 0.05 value in this study as also mentioned Materials and Methods section.

  1. Figure 6. From QQ plot it seems that the reported significant SNPs could be type I error. I therefore suggest to add Li and Ji (2005) or similar threshold to refine the significant SNPs.

Answer: It was observed that p values showed less deviation from the expected p values. The blue horizontal line designates the threshold (P< 0.0001) of significance, after crossing the false discovery rate (FDR) at 0.05 using the Bonferroni adjustment.

  1. Line 333- too long results. Make it short.

Answer: In revised manuscript, we have concise the results.

  1. Poor discussion, I suggest to reform this part. First two sentences are too broad and not justify the important of this study.

Answer: Thanks for the critical review: we have revised the discussion section carefully.

  • Try to come up the essence of the study, do not simply mention your previous findings. How many of those candidate genes were validated in other research, how many of them are regulating the pathways for determining the traits.

Answer:  Thanks for the improvement. We have revised the conclusion.

Thank you very much for your time and consideration.

Sincerely

Dr. Muhammad Ashfaq,

Associate Professor

Department of Plant Breeding & Genetics,

Faculty of Agricultural Sciences, University of the Punjab, Lahore.

Reviewer 3 Report

I study the article Quick genome-wide screening for yield and yield related traits in rice (Oryza Sativa L.) using SNPs markers. The article is well organized. Only a few things need to be answered in the article.

Write all terms in full for the first time and write them in short form every time. For example, you wrote Genome-wide association study (GWAS) in both line 47 and line 65. There is a lot of disorganization in this regard in the article

Why did you use a Randomized Complete Block Design?

Did you use the SES system to measure traits? to be mentioned

Many articles have now been published in this field. In my opinion, in order to make them more up-to-date, it is better to draw relevant gene networks in important areas with bioinformatic software to distinguish this article from other articles.

Author Response

Dear Editors of Genes, MDPI,

Thank you very much for your constructive comments regarding our manuscript entitled "Quick genome-wide screening for yield and yield related traits in rice (Oryza Sativa L.) using SNPs markers”. We have carefully revised the manuscript according to the suggestions. Our point-by-point responses to the Reviewers comments are presented below.

Reviewer 3:

  1. I study the article Quick genome-wide screening for yield and yield related traits in rice (Oryza Sativa L.) using SNPs markers. The article is well organized. Only a few things need to be answered in the article.

Answer: Thanks for the positive evaluation.

  1. Write all terms in full for the first time and write them in short form every time. For example, you wrote Genome-wide association study (GWAS) in both line 47 and line 65.There is a lot of disorganization in this regard in the article

Answer: Thanks for the nice suggestion. We have carefully revised the manuscript in this regard.

  1. Why did you use a Randomized Complete Block Design?

Answer: The randomized complete block design (RCBD) is a standard design for agricultural experiments in which similar experimental units are grouped into blocks or replicates. It is used to control variation in an experiment. We used the RCBD as it accounts the spatial effects in the field condition e.g variation in soil fertility level etc.

  1. Did you use the SES system to measure traits? to be mentioned

Answer: Thanks for the inquiry. No, we have only used conventional methods for the traits measurement.

  1. Many articles have now been published in this field. In my opinion, in order to make them more up-to-date, it is better to draw relevant gene networks in important areas with bioinformatic software to distinguish this article from other articles.

Answer: Thanks for the nice suggestion. We have already provided the table showing the candidate gene identification. In our opinion it’s sufficient to show the results in tabulated format as other studies also used tabulated format to show the results. we could not find such article which showed the network of genes identified by GWAS. If So, you have some reference article, it would be a great help if you can provide us some reference so that we can follow the method.

Thank you very much for your time and consideration.

Sincerely

Dr. Muhammad Ashfaq,

Associate Professor

Department of Plant Breeding & Genetics,

Faculty of Agricultural Sciences, University of the Punjab, Lahore.

Round 2

Reviewer 2 Report

Dear Authors,

It is a crucial research, but there are plenty of scopes to improve the write-up and presentation of the manuscript. Many comments are made throughout the text.

I suggest revising the title. It is a regular Genome-wide screening, not a ‘Quick genome-wide screening…’ which is quicker as you explained over bi-parental or multi-parental mapping populations.

 The abstract needs to be precise, informative, and to the point. Several of my comments should help.

In the Introduction, work on paragraph by paragraph, one at a time, and please organize your paragraphs for better flow and clarity.

Line 185 citation did not match MLM, the cited article was a famous genotyping by sequencing protocol paper. Please check this citation and carefully check rests.

The major flaws were not addressed. There are several inconsistencies remaining throughout the text (Please see comments in Line 222, 227, 294 and beyond).

Checking heritability would not alter GWAS results; instead, it could support GWAS analysis that it probably did not capture noise while mapping loci. I would not suggest adding heritability if you obtained clear peaks and straightforward Q-Q plots. It is expected that the SNPs on the upper right section of the graph deviate from the diagonal (true association, check GAPIT manusal). None of your Figure 6 QQ-plots exhibits such associations. The paper you forwarded is a great example.

Please look at the clear peaks in the Manhattan plot and QQ plot (right side), the SNPs on the upper right section of the QQ plot are depicting true association (Volante et al. 2020, Figure 2 a). As your traits are more complex and you did not obtain such true associated SNPs in QQ plot, you need to examine heritability critically for the evaluated traits and present in the manuscript. 

For your information, in your response you mentioned you added Bonferroni adjusted if so then threshold should be 10-5 if you used 7098 markers (0.05/7098= 7.044238e-06, -log10(7.044238e-06) = 5.152166, LOD score of 5 or 10-5), not 10-4.

Line 351, Figure 4 (b), it should be genome wide linkage disequilibrium (LD) (please see the comments Line 351) but the decay pattern is not clear. For your benefit I am adding one expected Figure here (Supplementary Figure 7, Bari et al, 2023, The Plant Phenome Journal).

And for chromosome wise LD then you should have 10 Figures as in rice there are 10 haploid chromosomes. You can explore chromosomes wise LD in Gali et al. 2019, Frontiers in Plant Science, Figure 2) just to visualize.

Please go through my comments and beyond and improve your nice piece of work so it can add value to the Science.

Thank you,

Reviewer

Author Response

Dear Editors of Genes, MDPI,

Thank you very much for your constructive comments regarding our manuscript entitled " Genome-wide association mapping for yield and yield related traits in rice (Oryza Sativa L.) using SNPs markers”. We have carefully revised the manuscript according to the suggestions. Our point-by-point responses to the Reviewers comments are presented below.

  1. It is a crucial research, but there are plenty of scopes to improve the write-up and presentation of the manuscript. Many comments are made throughout the text.

Answer: Thanks for your constructive comments. All the comments are addressed and please see the improvements in yellow highlights.

  1. I suggest revising the title. It is a regular Genome-wide screening, not a ‘Quick genome-wide screening…’ which is quicker as you explained over bi-parental or multi-parental mapping populations.

Answer: Thanks for your nice suggestion. We have deleted the word “quick”.  The revised title is as follow: Genome-wide association mapping for yield and yield-related traits in rice (Oryza Sativa L.) using SNPs markers.

  1. The abstract needs to be precise, informative, and to the point. Several of my comments should help.

Answer: Thanks for the constructive comments. We have carefully addressed all the comments and makes improvements based on the comments and please see the yellow highlights. 

  1. In the Introduction, work on paragraph by paragraph, one at a time, and please organize your paragraphs for better flow and clarity.

Answer: We have reorganized the introduction section. Please see the changes in track change system and yellow highlights.

  1. Line 185 citation did not match MLM, the cited article was a famous genotyping by sequencing protocol paper. Please check this citation and carefully check rests.

Answer: Thanks for pointing this out. We have replaced the reference with most relevant reference. Please see the line 165 and reference number 20.

  1. The major flaws were not addressed. There are several inconsistencies remaining throughout the text (Please see comments in Line 222, 227, 294 and beyond).

Answers: Thanks for pointing this out. Several your comments have already been address. Some remaining comments are as follow:

  1. You asked to move the Figure 1 in supplementary file. But we would like to keep in the main file. It is requested to please consider as it is.
  2. Based on your suggestion Table 1 (PCA table) is moved to supplementary file.
  • Based on your suggestion the information of markers is added in line 165.
  1. Added the references in line 210.
  2. Checking heritability would not alter GWAS results; instead, it could support GWAS analysis that it probably did not capture noise while mapping loci. I would not suggest adding heritability if you obtained clear peaks and straightforward Q-Q plots. It is expected that the SNPs on the upper right section of the graph deviate from the diagonal (true association, check GAPIT manusal). None of your Figure 6 QQ-plots exhibits such associations. The paper you forwarded is a great example.

Please look at the clear peaks in the Manhattan plot and QQ plot (right side), the SNPs on the upper right section of the QQ plot are depicting true association (Volante et al. 2020, Figure 2 a). As your traits are more complex and you did not obtain such true associated SNPs in QQ plot, you need to examine heritability critically for the evaluated traits and present in the manuscript. 

Answer: Thanks for nice suggestions. For the validation of GWAS results, the population structure and false discovery rate was calculated. In our opinion if we add heritability it will change the entire results of GWAS. It’s not always necessary to calculate the heritability to validate the GWAS results. There are many papers without the heritability calculation. Please see (https://doi.org/10.1186/s12284-020-00431-2). So, it is requested that please consider our previous results. It was observed that p values showed less deviation from the expected p values. The blue horizontal line designates the threshold (P< 0.0001) of significance, after crossing the false discovery rate (FDR) at 0.05 using the Bonferroni adjustment. In our study, the peak values are acceptable, regarding this the article has already been published. It is requested that please consider and accept our results for further proceedings.

  1. For your information, in your response you mentioned you added Bonferroni adjusted if so then threshold should be 10-5 if you used 7098 markers (0.05/7098= 7.044238e-06, -log10(7.044238e-06) = 5.152166, LOD score of 5 or 10-5), not 10-4.

Answer: Thanks for nice suggestion, as per your instruction the information has been added in the main file. The threshold for describing a marker to be significant was taken at 10-5 or above after crossing the false discovery rate (FDR) at 0.05 value to avoid the False marker trait association.

  1. Line 351, Figure 4 (b), it should be genome wide linkage disequilibrium (LD) (please see the comments Line 351) but the decay pattern is not clear. For your benefit I am adding one expected Figure here (Supplementary Figure 7, Bari et al, 2023, The Plant Phenome Journal).

And for chromosome wise LD then you should have 10 Figures as in rice there are 10 haploid chromosomes. You can explore chromosomes wise LD in Gali et al. 2019, Frontiers in Plant Science, Figure 2) just to visualize.

Answer: Thanks for critical review. The QTL intervals were limited to regions where the r2 values (squared allele frequency correlation) between markers were above 0.4 (P < 1e-04). In case the observed LD block around significant marker(s) was less than 50 kb, we extended the QTLs up to 50 kb upstream and downstream of the detected regions. In this study, GWAS result obtained from GAPIT and Figure 4 are again generated, and their detail included in the revised manuscript like “Each dot represents a pair of distances between two markers on the window and their squared correlation coefficient. The red line is the moving average of the 10 adjacent markers.

Please go through my comments and beyond and improve your nice piece of work so it can add value to the Science.

Sincerely

Dr. Muhammad Ashfaq,

Associate Professor

Department of Plant Breeding & Genetics,

Faculty of Agricultural Sciences, University of the Punjab, Lahore.
